# FANTASTIC FOUR: DIFFERENTIABLE BOUNDS ON SINGULAR VALUES OF CONVOLUTION LAYERS

**Sahil Singla & Soheil Feizi**
Department of Computer Science
University of Maryland
College Park, MD 20740, USA
{ssingla,sfeizi}@umd.edu

## ABSTRACT

In deep neural networks, the spectral norm of the Jacobian of a layer bounds the factor by which the norm of a signal changes during forward/backward propagation. Spectral norm regularizations have been shown to improve generalization, robustness and optimization of deep learning methods. Existing methods to compute the spectral norm of convolution layers either rely on heuristics that are efficient in computation but lack guarantees or are theoretically-sound but computationally expensive. In this work, we obtain the best of both worlds by deriving *four* provable upper bounds on the spectral norm of a standard 2D multi-channel convolution layer. These bounds are differentiable and can be computed efficiently during training with negligible overhead. One of these bounds is in fact the popular heuristic method of Miyato et al. (2018) (multiplied by a constant factor depending on filter sizes). Each of these four bounds can achieve the tightest gap depending on convolution filters. Thus, we propose to use the minimum of these four bounds as a tight, differentiable and efficient upper bound on the spectral norm of convolution layers. We show that our spectral bound is an effective regularizer and can be used to bound either the lipschitz constant or curvature values (eigenvalues of the Hessian) of neural networks. Through experiments on MNIST and CIFAR-10, we demonstrate the effectiveness of our spectral bound in improving generalization and provable robustness of deep networks.

## 1 INTRODUCTION

Bounding singular values of different layers of a neural network is a way to control the complexity of the model and has been used in different problems including robustness, generalization, optimization, generative modeling, etc. In particular, the spectral norm (the maximum singular value) of a layer bounds the factor by which the norm of the signal increases or decreases during both forward and backward propagation within that layer. If all singular values are all close to one, then the gradients neither explode nor vanish (Hochreiter, 1991; Hochreiter et al., 2001; Klambauer et al., 2017; Xiao et al., 2018). Spectral norm regularizations/bounds have been used in improving the generalization (Bartlett et al., 2017; Long & Sedghi, 2020), in training deep generative models (Arjovsky et al., 2017; Gulrajani et al., 2017; Tolstikhin et al., 2018; Miyato et al., 2018; Hoogeboom et al., 2020) and in robustifying models against adversarial attacks (Singla & Feizi, 2020; Szegedy et al., 2014; Peck et al., 2017; Zhang et al., 2018; Anil et al., 2018; Hein & Andriushchenko, 2017; Cisse et al., 2017). These applications have resulted in multiple works to regularize neural networks by penalizing the spectral norm of the network layers (Drucker & Le Cun, 1992; Yoshida & Miyato, 2017; Miyato et al., 2018; 2017; Sedghi et al., 2019; Singla & Feizi, 2020).

For a fully connected layer with weights $\mathbf{W}$ and bias $\mathbf{b}$, the lipschitz constant is given by the spectral norm of the weight matrix i.e, $\|\mathbf{W}\|_2$, which can be computed efficiently using the power iteration method (Golub & Van Loan, 1996). In particular, if the matrix $\mathbf{W}$ is of size $p \times q$, the computational complexity of power iteration (assuming convergence in constant number of steps) is $\mathcal{O}(pq)$.

Convolution layers (Lecun et al., 1998) are one of the key components of modern neural networks, particularly in computer vision (Krizhevsky et al., 2012). Consider a convolution filter $\mathbf{L}$ of size

$c_{out} \times c_{in} \times h \times w$ where $c_{out}$, $c_{in}$, $h$ and $w$ denote the number of output channels, input channels, height and width of the filter respectively; and a square input sample of size $c_{in} \times n \times n$ where $n$ is its height and width. A naive representation of the Jacobian of this layer will result in a matrix of size $n^2 c_{out} \times n^2 c_{in}$. For a typical convolution layer with the filter size $64 \times 3 \times 7 \times 7$ and an ImageNet sized input $3 \times 224 \times 224$ (Krizhevsky et al., 2012), the corresponding jacobian matrix has a very large size: $802816 \times 150528$. This makes an explicit computation of the jacobian infeasible. Ryu et al. (2019) provide a way to compute the spectral norm of convolution layers using convolution and transposed convolution operations in power iteration, thereby avoiding this explicit computation. This leads to an improved running time especially when the number of input/output channels is small (Table 1).

However, in addition to the running time, there is an additional difficulty in the approach proposed in Ryu et al. (2019) (and other existing approaches described later) regarding the computation of the spectral norm gradient (often used as a regularization during the training). The gradient of the largest singular value with respect to the jacobian can be naively computed by taking the outer product of corresponding singular vectors. However, due to the special structure of the convolution operation, the jacobian will be a sparse matrix with repeated elements (see Appendix Section D for details). The naive computation of the gradient will result in non-zero gradient values at elements that should be in fact zeros throughout training and also will assign different gradient values at elements that should always be identical. These issues make the gradient computation of the spectral norm with respect to the convolution filter weights using the technique of Ryu et al. (2019) difficult.

Recently, Sedghi et al. (2019) provided a principled approach for exactly computing the singular values of convolution layers. They construct $n^2$ matrices each of size $c_{out} \times c_{in}$ by taking the Fourier transform of the convolution filter (details in Appendix Section B). The set of singular values of the jacobian equals the union of singular values of these $n^2$ matrices. However, this method can have high computational complexity since it requires SVD of $n^2$ matrices. Although this method can be adapted to compute the spectral norm of $n^2$ matrices using power iteration (in parallel with a GPU implementation) instead of full SVD, the intrinsic computational complexity (discussed in Table 2) can make it difficult to use this approach for very deep networks and large input sizes especially when computational resources are limited. Moreover, computing the gradient of the spectral norm using this method is not straightforward since each of these $n^2$ matrices contain complex numbers. Thus, Sedghi et al. (2019) suggests to clip the singular values if they are above a certain threshold to bound the spectral norm of the layer. In order to reduce the training overhead, they clip the singular values only after every 100 iterations. The resulting method reduces the training overhead but is still costly for large input sizes and very deep networks. We report the running time of this method in Table 1 and its training time for one epoch (using 1 GPU implementation) in Table 4c.

Because of the aforementioned issues, efficient methods to control the spectral norm of convolution layers have resorted to heuristics (Yoshida & Miyato, 2017; Miyato et al., 2018; Gouk et al., 2018). Typically, these methods reshape the convolution filter of dimensions $c_{out} \times c_{in} \times h \times w$ to construct a matrix of dimensions $c_{out} \times hwc_{in}$, and use the spectral norm of this matrix as an estimate of the spectral norm of the convolution layer. To regularize during training, they use the outer product of the corresponding singular vectors as the gradient of the largest singular value with respect to the reshaped matrix. Since the weights do not change significantly during each training step, they use only one iteration of power method during each step to update the singular values and vectors (using the singular vectors computed in the previous step). These methods result in negligible overhead during the training. However, due to lack of theoretical justifications (which we resolve in this work), they are not guaranteed to work for all different shapes and weights of the convolution filter. Previous studies have observed under estimation of the spectral norm using these heuristics (Jiang et al., 2019).

On one hand, there are computationally efficient but heuristic ways of computing and bounding the spectral norm of convolutional layers (Miyato et al., 2017; 2018). On the other hand, the exact computation of the spectral norm of convolutional layers proposed by Sedghi et al. (2019); Ryu et al. (2019) can be expensive for commonly used architectures especially with large inputs such as ImageNet samples. Moreover, the difficulty in computing the gradient of the spectral norm with respect to the jacobian under these methods make their use as regularization during the training process challenging.

In this paper, we resolve these issues by deriving a differentiable and efficient *upper bound* on the spectral norm of convolutional layers. Our bound is provable and not based on heuristics. Our

| Filter shape | Spectral norm bounds | | | | | | Running time (secs) | | |
| | $\sqrt{hw} \times$ | | | | **Bound (Ours)** | Exact (Sedghi / Ryu) | **Ours** | Sed-ghi | Ryu |
| | $\|\mathbf{R}\|_2$ | $\|\mathbf{S}\|_2$ | $\|\mathbf{T}\|_2$ | $\|\mathbf{U}\|_2$ | | | | | |
|---|---|---|---|---|---|---|---|---|---|
| $64 \times 3 \times 7 \times 7$ | 47.29 | 46.37 | **28.89** | 77.31 | **28.89** | 15.92 | 0.004 | 0.94 | 0.03 |
| $64 \times 64 \times 3 \times 3$ | 9.54 | 9.61 | **9.33** | 10.51 | **9.33** | 6.01 | 0.033 | 8.46 | 0.04 |
| $64 \times 64 \times 3 \times 3$ | 6.57 | **6.30** | 6.49 | 7.86 | **6.30** | 5.34 | 0.033 | 9.19 | 0.04 |
| $64 \times 64 \times 3 \times 3$ | 8.92 | **8.71** | 8.80 | 10.97 | **8.71** | 7.00 | 0.033 | 9.16 | 0.04 |
| $64 \times 64 \times 3 \times 3$ | **5.40** | 5.52 | 5.81 | 6.30 | **5.40** | 3.82 | 0.033 | 9.25 | 0.03 |
| $128 \times 64 \times 3 \times 3$ | 6.99 | 6.92 | **6.01** | 9.12 | **6.01** | 4.71 | 0.033 | 3.08 | 0.05 |
| $128 \times 128 \times 3 \times 3$ | 7.45 | 7.38 | **7.21** | 8.51 | **7.21** | 5.72 | 0.033 | 8.98 | 0.08 |
| $128 \times 128 \times 3 \times 3$ | **6.78** | 7.18 | 6.85 | 8.78 | **6.78** | 4.41 | 0.032 | 9.02 | 0.08 |
| $128 \times 128 \times 3 \times 3$ | **7.57** | 7.72 | 8.89 | 7.87 | **7.57** | 4.88 | 0.033 | 9.00 | 0.08 |
| $256 \times 128 \times 3 \times 3$ | 8.58 | **8.45** | 8.67 | 8.77 | **8.45** | 7.39 | 0.032 | 3.92 | 0.14 |
| $256 \times 256 \times 3 \times 3$ | 8.19 | 8.07 | **8.04** | 8.99 | **8.04** | 6.58 | 0.033 | 13.1 | 0.26 |
| $256 \times 256 \times 3 \times 3$ | 8.06 | 8.13 | **7.57** | 9.26 | **7.57** | 6.36 | 0.034 | 11.2 | 0.26 |
| $256 \times 256 \times 3 \times 3$ | 9.91 | 9.76 | 10.97 | **9.17** | **9.17** | 7.68 | 0.033 | 11.2 | 0.26 |
| $512 \times 256 \times 3 \times 3$ | 11.24 | **11.00** | 11.50 | 11.07 | **11.00** | 9.99 | 0.034 | 5.26 | 0.51 |
| $512 \times 512 \times 3 \times 3$ | 10.98 | 10.65 | 11.91 | **10.45** | **10.45** | 9.09 | 0.033 | 15.7 | 1.04 |
| $512 \times 512 \times 3 \times 3$ | 20.22 | 19.90 | 21.91 | **18.37** | **18.37** | 17.60 | 0.033 | 15.7 | 1.04 |
| $512 \times 512 \times 3 \times 3$ | 7.83 | 7.99 | 8.26 | **7.60** | **7.60** | 7.48 | 0.034 | 15.8 | 1.04 |

Table 1: Comparison between the exact spectral norm of the jacobian of convolution layers (computed using Sedghi et al. (2019); Ryu et al. (2019)) and our proposed bound for the a Resnet-18 network pre-trained on ImageNet. Our bound is within 1.5 times the exact spectral norm (except the first layer) while being significantly faster to compute compared to the exact methods. All running times were computed on GPU using tensorflow. For Sedghi et al. (2019)'s method, to have a fair comparison, we *only* compute the largest singular value (and not all singular values) for all individual matrices in parallel using power iteration (on 1 GPU) and take the maximum. We observe that for different filters, different components of our bound give the minimum value. Also, the values of the four bounds can be very different for different convolution filters (example: filter in the first layer).

computational complexity is similar to that of heuristics (Miyato et al., 2017; 2018) allowing our bound to be used as a regularizer for efficiently training deep convolutional networks. In this way, our proposed approach combines the benefits of the speed of the heuristics and the theoretical rigor of Sedghi et al. (2019). Table 2 summarizes the differences between previous works and our approach. In Table 1, we empirically observe that our bound can be computed in a time significantly faster than Sedghi et al. (2019); Ryu et al. (2019), while providing a guaranteed upper bound on the spectral norm. Moreover, we empirically observe that our upper bound and the exact value are close to each other (Section 3.1).

Below, we briefly explain our main result. Consider a convolution filter $\mathbf{L}$ of dimensions $c_{out} \times c_{in} \times h \times w$ and input of size $c_{in} \times n \times n$. The corresponding jacobian matrix $\mathbf{J}$ is of size $n^2 c_{out} \times n^2 c_{in}$. We show that the largest singular value of the jacobian (i.e. $\|\mathbf{J}\|_2$) is bounded as:

$$\|\mathbf{J}\|_2 \le \sqrt{hw} \min \left( \|\mathbf{R}\|_2, \|\mathbf{S}\|_2, \|\mathbf{T}\|_2, \|\mathbf{U}\|_2 \right),$$

where $\mathbf{R}, \mathbf{S}, \mathbf{T}$ and $\mathbf{U}$ are matrices of sizes $hc_{out} \times wc_{in}$, $wc_{out} \times hc_{in}$, $c_{out} \times hwc_{in}$ and $hwc_{out} \times c_{in}$ respectively, and can be computed by appropriately reshaping the filter $\mathbf{L}$ (details in Section 3). This upper bound is independent of the input width and height ($n$). Formal results are stated in Theorem 1 and proved in the appendix. Remarkably, $\|\mathbf{T}\|_2$ is the heuristic suggested by Miyato et al. (2018). To the best of our knowledge, this is the first work that derives a provable bound on the spectral norm of convolution filter as a constant factor (dependant on filter sizes, but not filter weights) times the heuristic of Miyato et al. (2018). In Tables 1 and 3, we show that the other 3 bounds (using

$\|\mathbf{R}\|_2, \|\mathbf{S}\|_2, \|\mathbf{U}\|_2$) can be significantly smaller than $\sqrt{hw}\|\mathbf{T}\|_2$ for different convolution filters. Thus, we take the minimum of these 4 quantities to bound the spectral norm of a convolution filter.

In Section 4, we show that our bound can be used to improve the generalization and robustness properties of neural networks. Specifically, we show that using our bound as a regularizer during training, we can achieve improvement in accuracy on par with exact method (Sedghi et al., 2019) while being significantly faster to train (Table 4). We also achieve significantly higher robustness certificates against adversarial attacks than CNN-Cert (Boopathy et al., 2018) on a single layer CNN (Table 5). These results demonstrate potentials for practical uses of our results. Code is available at the github repository: https://github.com/singlasahil14/fantastic-four.

| | Exact (Sedghi et al., 2019) | Exact (Ryu et al., 2019) | **Upper Bound (Ours)** |
|---|---|---|---|
| Computation | Norm of $n^2$ matrices, each of size: $c_{out}c_{in}$ | Norm of one matrix of size: $n^4 c_{out}c_{in}$ | Norm of four matrices, each of size: $c_{out}c_{in}hw$ |
| Time complexity ($\mathcal{O}$) | $n^2\, c_{out}\, c_{in}$ | $n^2\, h\, w\, c_{out}\, c_{in}$ | $h\, w\, c_{out}\, c_{in}$ |
| Guaranteed bound | ✓ | ✓ | ✓ |
| Easy gradient computation | ✗ | ✗ | ✓ |

Table 2: Comparison of various methods used for computing the norm of convolution layers. $n$ is the height and width for a square input, $c_{in}$ is the number of input channels, $c_{out}$ is the number of output channels, $h$ and $w$ are the height and width of the convolution filter. For Sedghi et al. (2019), we only compute the largest singular value using power iteration (i.e not all singular values).

## 2 NOTATION

For a vector $\mathbf{v}$, we use $\mathbf{v}_j$ to denote the element in the $j^{th}$ position of the vector. We use $\mathbf{A}_{j,:}$ and $\mathbf{A}_{:,k}$ to denote the $j^{th}$ row and $k^{th}$ column of the matrix $\mathbf{A}$ respectively. We assume both $\mathbf{A}_{j,:}$, $\mathbf{A}_{:,k}$ to be column vectors (thus $\mathbf{A}_{j,:}$ is the transpose of $j^{th}$ row of $\mathbf{A}$). $\mathbf{A}_{j,k}$ denotes the element in $j^{th}$ row and $k^{th}$ column of $\mathbf{A}$. The same rules can be directly extended to higher order tensors. For a matrix $\mathbf{A} \in \mathbb{R}^{q \times r}$ and a tensor $\mathbf{B} \in \mathbb{R}^{p \times q \times r}$, $vec(\mathbf{A})$ denotes the vector constructed by stacking the rows of $\mathbf{A}$ and $vec(\mathbf{B})$ denotes the vector constructed by stacking the vectors $vec(\mathbf{B}_{j,:,:})$, $j \in [p-1]$:

$$vec(\mathbf{A})^T = \begin{bmatrix} \mathbf{A}_{0,:}^T, & \mathbf{A}_{1,:}^T, & \cdots, & \mathbf{A}_{q-1,:}^T \end{bmatrix} \qquad vec(\mathbf{B})^T = \begin{bmatrix} \mathbf{B}_{0,:,:}^T, & \mathbf{B}_{1,:,:}^T, & \cdots, & \mathbf{B}_{p-1,:,:}^T \end{bmatrix}$$

We use the following notation for a convolutional neural network. $L$ denotes the number of layers and $\phi$ is the activation function. For an input $\mathbf{x}$, we use $\mathbf{z}^{(I)}(\mathbf{x}) \in \mathbf{R}^{N_I}$ and $\mathbf{a}^{(I)}(\mathbf{x}) \in \mathbf{R}^{N_I}$ to denote the raw (*before* applying $\phi$) and activated (*after* applying $\phi$) neurons in the $I^{th}$ hidden layer respectively. $\mathbf{a}^{(0)}$ denotes the input image $\mathbf{x}$. To simplify notation and when no confusion arises, we make the dependency of $\mathbf{z}^{(I)}$ and $\mathbf{a}^{(I)}$ to $\mathbf{x}$ implicit. $\phi'(\mathbf{z}^{(I)})$ and $\phi''(\mathbf{z}^{(I)})$ denotes the elementwise first and second derivatives of $\phi$ at $\mathbf{z}^{(1)}$. $\mathbf{W}^{(I)}$ denotes the weights for the $I^{th}$ layer i.e $\mathbf{W}^{(I)}$ will be a tensor for a convolution layer and a matrix for a fully connected layer. $\mathbf{J}^{(I)}$ denotes the jacobian matrix of $vec(\mathbf{z}^{(1)})$ with respect to the input $\mathbf{x}$. $\theta$ denotes the neural network parameters. $f_\theta(\mathbf{x})$ denotes the softmax probabilities output by the network for an input $\mathbf{x}$. For an input $\mathbf{x}$ and label $y$, the cross entropy loss is denoted by $\ell\big(f_\theta(\mathbf{x}), y\big)$.

## 3 MAIN RESULTS

Consider a convolution filter $\mathbf{L}$ of size $c_{out} \times c_{in} \times h \times w$ applied to an input $\mathbf{X}$ of size $c_{in} \times n \times n$. The filter $\mathbf{L}$ takes an input patch of size $c_{in} \times h \times w$ from the input and outputs a vector of size $c_{out}$ for every such patch. The same operation is applied across all such patches in the image. To apply convolution at the edges of the image, modern convolution layers either not compute the outputs

thus reducing the size of the generated feature map, or pad the input with zeros to preserve its size. When we pad the image with zeroes, the corresponding jacobian becomes a *toeplitz matrix*. Another version of convolution treats the input as if it were a torus; when the convolution operation calls for a pixel off the right end of the image, the layer "wraps around" to take it from the left edge, and similarly for the other edges. For this version of convolution, the jacobian is a *circulant matrix*. The quality of approximations between toeplitz and circulant colvolutions has been analyzed in the case of 1D (Gray, 2005) and 2D signals (Zhu & Wakin, 2017). For the 2D case (similar to the 1D case), $O(1/p)$ bound is obtained on the error, where $p \times p$ is the size of both (topelitz and circulant) matrices. Consequently, theoretical analysis of convolutions that wrap around the input (i.e using circulant matrices) has been become standard. This is the case that we analyze in this work. Furthermore, we assume that the stride size to be 1 in both horizontal and vertical directions.

The output $\mathbf{Y}$ produced from applying the filter $\mathbf{L}$ to an input $\mathbf{X}$ is of size $c_{out} \times n \times n$. The corresponding jacobian ($\mathbf{J}$) will be a matrix of size $n^2 c_{out} \times n^2 c_{in}$ satisfying:

$$vec(\mathbf{Y}) = \mathbf{J} vec(\mathbf{X}).$$

Our goal is to bound the norm of jacobian of the convolution operation i.e, $\|\mathbf{J}\|_2$. Sedghi et al. (2019) also derive an expression for the exact singular values of the jacobian of convolution layers. However, their method requires the computation of the spectral norm of $n^2$ matrices (each matrix of size $c_{out} \times c_{in}$) for every convolution layer. We extend their result to derive a differentiable and easy-to-compute upper bound on singular values stated in the following theorem:

**Theorem 1.** *Consider a convolution filter $\mathbf{L}$ of size $c_{out} \times c_{in} \times h \times w$ that applied to input $\mathbf{X}$ of size $c_{in} \times n \times n$ gives output $\mathbf{Y}$ of size $c_{out} \times n \times n$. The jacobian of $\mathbf{Y}$ with respect to $\mathbf{X}$ (i.e. $\mathbf{J}$) will be a matrix of size $n^2 c_{out} \times n^2 c_{in}$. The spectral norm of $\mathbf{J}$ is bounded by:*

$$\|\mathbf{J}\|_2 \leq \sqrt{hw} \, \min\left(\|\mathbf{R}\|_2, \|\mathbf{S}\|_2, \|\mathbf{T}\|_2, \|\mathbf{U}\|_2\right),$$

*where the matrices $\mathbf{R}, \mathbf{S}, \mathbf{T}$ and $\mathbf{U}$ are defined as follows:*

$$\mathbf{R} = \begin{bmatrix} \mathbf{L}_{:,:,0,0} & \mathbf{L}_{:,:,0,1} & \cdots & \mathbf{L}_{:,:,0,w-1} \\ \mathbf{L}_{:,:,1,0} & \mathbf{L}_{:,:,1,1} & \cdots & \mathbf{L}_{:,:,1,w-1} \\ \vdots & \vdots & \ddots & \vdots \\ \mathbf{L}_{:,:,h-1,0} & \mathbf{L}_{:,:,h-1,1} & \cdots & \mathbf{L}_{:,:,h-1,w-1} \end{bmatrix}, \quad \mathbf{S} = \begin{bmatrix} \mathbf{L}_{:,:,0,0} & \mathbf{L}_{:,:,1,0} & \cdots & \mathbf{L}_{:,:,h-1,0} \\ \mathbf{L}_{:,:,0,1} & \mathbf{L}_{:,:,1,1} & \cdots & \mathbf{L}_{:,:,h-1,1} \\ \vdots & \vdots & \ddots & \vdots \\ \mathbf{L}_{:,:,0,w-1} & \mathbf{L}_{:,:,1,w-1} & \cdots & \mathbf{L}_{:,:,h-1,w-1} \end{bmatrix}$$

$$\mathbf{T} = \begin{bmatrix} \mathbf{A}_0 & \mathbf{A}_1 & \cdots & \mathbf{A}_{h-1} \end{bmatrix}, \quad where \quad \mathbf{A}_l = \begin{bmatrix} \mathbf{L}_{:,:,l,0} & \mathbf{L}_{:,:,l,1} & \cdots & \mathbf{L}_{:,:,l,w-1} \end{bmatrix}$$

$$\mathbf{U}^T = \begin{bmatrix} \mathbf{B}_0^T & \mathbf{B}_1^T & \cdots & \mathbf{B}_{h-1}^T \end{bmatrix}, \quad where \quad \mathbf{B}_l^T = \begin{bmatrix} \mathbf{L}_{:,:,l,0}^T & \mathbf{L}_{:,:,l,1}^T & \cdots & \mathbf{L}_{:,:,l,w-1}^T \end{bmatrix}$$

Proof of Theorem 1 is in Appendix E. The matrices $\mathbf{R}, \mathbf{S}, \mathbf{T}$ and $\mathbf{U}$ are of dimensions $c_{out}h \times c_{in}w, c_{out}w \times c_{in}h, c_{out} \times c_{in}hw$ and $c_{out}hw \times c_{in}$ respectively. In the current literature (Miyato et al., 2018), the heuristic used for estimating the spectral norm involves combining the dimensions of sizes $h, w, c_{in}$ in the filter $\mathbf{L}$ to create the matrix $\mathbf{T}$ of dimensions $c_{out} \times hwc_{in}$. The norm of resulting matrix is used as a heuristic estimate of the spectral norm of the jacobian of convolution operator. However, in Theorem 1 we show that the norm of this matrix multiplied with a factor of $\sqrt{hw}$ gives a provable upper bound on the singular values of the jacobian. In Tables 1 and 3, we show that for different convolution filters, there can be significant differences between the four bounds and any of these four bounds can be the minimum.

### 3.1 TIGHTNESS ANALYSIS

In Appendix F, we show that the bound is exact for a convolution filter with $h = w = 1$:

**Lemma 1.** *For $h = 1, w = 1$, the bounds in Theorem 1 are exact i.e:*

$$\|\mathbf{J}\|_2 = \|\mathbf{R}\|_2 = \|\mathbf{S}\|_2 = \|\mathbf{T}\|_2 = \|\mathbf{U}\|_2$$

In Table 3, we analyze the tightness between our bound and the exact largest singular value computed by Sedghi et al. (2019) for different filter shapes. Each convolution filter was constructed by sampling from a standard normal distribution $\mathcal{N}(0,1)$. We observe that the bound is tight for small filter sizes but the ratio (Bound/Exact) becomes large for large $h$ and $w$. We also observe that the values computed by the four bounds can be significantly different and that we can get a significantly improved bound by taking the minimum of the four quantities. In Appendix Section G, Figure 1, we empirically observe that the gap between our upper bound and the exact value can become very small by adding our bound as a regularizer during training.

| Filter shape | $\sqrt{hw} \times$ | | | | Bound | Exact | Bound/ |
|---|---|---|---|---|---|---|---|
| | $\|\mathbf{R}\|_2$ | $\|\mathbf{S}\|_2$ | $\|\mathbf{T}\|_2$ | $\|\mathbf{U}\|_2$ | (Ours) | (Sedghi) | Exact |
| $64 \times 64 \times 2 \times 2$ | **44.19** | 46.11 | 46.76 | 47.88 | **44.19** | 31.96 | 1.38 |
| $64 \times 64 \times 3 \times 3$ | 84.03 | **82.71** | 95.27 | 97.36 | **82.71** | 48.77 | 1.70 |
| $64 \times 64 \times 5 \times 5$ | 179.09 | **177.31** | 237.22 | 238.30 | **177.31** | 82.39 | 2.15 |
| $64 \times 64 \times 7 \times 7$ | 297.02 | **296.18** | 444.69 | 451.52 | **296.18** | 116.05 | 2.55 |
| $64 \times 16 \times 2 \times 2$ | 32.27 | 32.45 | **31.18** | 38.20 | **31.18** | 23.25 | 1.34 |
| $64 \times 16 \times 3 \times 3$ | 60.74 | 62.42 | **59.67** | 82.17 | **59.67** | 37.09 | 1.61 |
| $64 \times 16 \times 5 \times 5$ | **130.07** | 132.62 | 136.16 | 216.22 | **130.07** | 61.01 | 2.13 |
| $64 \times 16 \times 7 \times 7$ | 220.25 | **220.20** | 248.50 | 415.14 | **220.20** | 86.54 | 2.54 |

Table 3: Tightness analysis between our proposed bound and exact method by Sedghi et al. (2019).

## 3.2 GRADIENT COMPUTATION

Since the matrix $\mathbf{R}$ can be directly computed by reshaping the filter weights $\mathbf{L}$ (equation 8), we can compute the derivative of our bound $\sqrt{hw}\|\mathbf{R}\|_2$ (or $\|\mathbf{S}\|_2$, $\|\mathbf{T}\|_2$, $\|\mathbf{U}\|_2$) with respect to filter weights $\mathbf{L}$ by first computing the derivative of $\sqrt{hw}\|\mathbf{R}\|_2$ with respect to $\mathbf{R}$ and then appropriately reshaping the obtained derivative.

Let $\mathbf{u}$ and $\mathbf{v}$ be the singular vectors corresponding to the largest singular value, i.e. $\|\mathbf{R}\|_2$. Then, the derivative of our upper bound $\sqrt{hw}\|\mathbf{R}\|_2$ with respect to $\mathbf{R}$ can be computed as follows:

$$\nabla_{\mathbf{R}} \sqrt{hw}\|\mathbf{R}\|_2 = \sqrt{hw}\,\mathbf{u}\mathbf{v}^T \quad \text{where } \|\mathbf{R}\|_2 = \mathbf{u}^T \mathbf{R} \mathbf{v}.$$

Moreover, since the weights do not change significantly during the training, we can use one iteration of power method to update $\mathbf{u}, \mathbf{v}$ and $\|\mathbf{R}\|_2$ during each training step (similar to Miyato et al. (2018; 2017)). This allows us to use our bound efficiently as a regularizer during the training process.

## 4 EXPERIMENTS

All experiments were conducted using a single NVIDIA GeForce RTX 2080 Ti GPU.

### 4.1 COMPARISON WITH EXISTING METHODS

In Table 1, we show a comparison between the exact spectral norms (computed using Sedghi et al. (2019), Ryu et al. (2019)) and our upper bound, i.e. $\sqrt{hw} \min(\|\mathbf{R}\|_2, \|\mathbf{S}\|_2, \|\mathbf{T}\|_2, \|\mathbf{U}\|_2)$ on a pre-trained Resnet-18 network (He et al., 2015). Except for the first layer, we observe that our bound is within 1.5 times the value of the exact spectral norm while being significantly faster to compute. Similar results can be observed in Table 3 for a standard gaussian filter. Thus, by taking the minimum of the four bounds, we can get a significant gain.

### 4.2 EFFECTS ON GENERALIZATION

In Table 4a, we study the effect of using our proposed bound as a training regularizer on the generalization error. We use a Resnet-32 neural network architecture and the CIFAR-10 dataset (Krizhevsky, 2009) for training. For regularization, we use the sum of spectral norms of all layers of the network during training. Thus, our regularized objective function is given as follows[1]:

$$\min_{\theta} \quad \mathbb{E}_{(\mathbf{x},y)} \left[ \ell\left(f_{\theta}(\mathbf{x}), y\right) \right] + \beta \sum_I u^{(I)} \tag{1}$$

where $\beta$ is the regularization coefficient, $(\mathbf{x}, y)$'s are the input-label pairs in the training data, $u^{(I)}$ denotes the bound for the $I^{th}$ convolution or fully connected layer. For the convolution layers, $u^{(I)}$

---

[1]We do not use the sum of log of spectral norm values since that would make the filter-size dependant factor of $\sqrt{hw}$ irrelevant for gradient computation.

is computed as $\sqrt{hw}\min(\|\mathbf{R}\|_2, \|\mathbf{S}\|_2, \|\mathbf{T}\|_2, \|\mathbf{U}\|_2)$ using Theorem 1. For fully connected layers, we can compute $u^{(I)}$ using power iteration (Miyato et al., 2018).

| $\beta$ | Test Accuracy | |
|---|---|---|
| | No weight decay | weight decay = $10^{-4}$ |
| 0 | 91.26% | 92.53% |
| 0.0008 | 91.62% | 92.37% |
| 0.0009 | 91.74% | 92.95% |
| 0.0010 | 91.79% | 93.13% |
| 0.0011 | 91.87% | 92.75% |
| 0.0012 | 91.81% | 92.53% |
| 0.0013 | 92.17% | 92.73% |
| 0.0014 | **92.23%** | 92.61% |
| 0.0015 | 91.87% | 93.15% |
| 0.0016 | 91.92% | **93.26%** |
| 0.0017 | 91.70% | 92.92% |
| 0.0018 | 91.52% | 91.89% |
| 0.0019 | 92.00% | 92.47% |
| 0.0020 | 91.82% | 92.56% |

(a) Our proposed regularizer

| Clipping threshold | Test Accuracy | |
|---|---|---|
| | No weight decay | weight decay = $10^{-4}$ |
| None | 91.26% | 92.53% |
| 0.1 | 90.67% | 91.87% |
| 0.5 | **92.31%** | **93.30%** |
| 1.0 | 91.92% | 92.86% |

(b) Singular value clipping (Sedghi et al., 2019)

| Method | Running time (s) | Increase (%) |
|---|---|---|
| Standard | 49.58 | – |
| **Ours** | 54.39 | 9.7% |
| Clipping (Sedghi et al., 2019) | 156.46 | 215.6% |

(c) Running time for 1 epoch on a dataset with 10,000 images of size 224x224x3

Table 4: Comparison between our proposed regularizer (a) and singular value clipping by Sedghi et al. (2019) (b) on test accuracy. Results are on CIFAR-10 dataset using a Resnet-32 network.

Since weight decay (Krogh & Hertz, 1991) indirectly minimizes the Frobenius norm squared which is equal to the sum of squares of singular values, it implicitly forces the largest singular values for each layer (i.e the spectral norm) to be small. Therefore, to measure the effect of our regularizer on the test set accuracy, we compare the effect of adding our regularizer both with and without weight decay. The weight decay coefficient was selected using grid search using 20 values between $[0, 2 \times 10^{-3}]$ using a held-out validation set of 5000 samples.

Our experimental results are reported in Table 4a. For the case of no weight decay, we observe an improvement of $0.97\%$ over the case when $\beta = 0$. When we include a weight decay of $10^{-4}$ in training, there is an improvement of $0.73\%$ over the baseline. Using the method of Sedghi et al. (2019) (i.e. clipping the singular values above 0.5 and 1) results in gain of $0.77\%$ (with weight decay) and $1.05\%$ (without weight decay) which is similar to the results mentioned in their paper. In addition to obtaining on par performance gains with the exact method of Sedghi et al. (2019), a key advantage of our approach is its very efficient running time allowing its use for very large input sizes and deep networks. We report the training time of these methods in Table 4(c) for a dataset with large input sizes. The increase in the training time using our method compared to the standard training is just $9.7\%$ while that for Sedghi et al. (2019) is around $215.6\%$.

## 4.3 EFFECTS ON PROVABLE ADVERSARIAL ROBUSTNESS

In this part, we show the usefulness of our spectral bound in enhancing provable robustness of convolutional classifiers against adversarial attacks (Szegedy et al., 2014). A robustness certificate is a lower bound on the minimum distance of a given input to the decision boundary of the classifier. For *any* perturbation of the input with a magnitude smaller than the robustness certificate value, the classification output will provably remain the same. However, computing exact robustness certificates requires solving a non-convex optimization, making the computation difficult for deep classifiers. In the last couple of years, several certifiable defenses against adversarial attacks have been proposed (e.g. Boopathy et al. (2018); Mirman et al. (2018); Zhang et al. (2018); Weng et al. (2018); Singla & Feizi (2020); Virmaux & Scaman (2018); Tsuzuku et al. (2018); Levine & Feizi (2020); Cohen

et al. (2019); Levine & Feizi (2020).) In particular, to show the usefulness of our spectral bound in this application, we examine the method proposed in Singla & Feizi (2020) that uses a bound on the lipschitz constant of network gradients (i.e. the curvature values of the network). Their robustness certification method in fact depends on the spectral norm of the Jacobian of different network layers; making a perfect case study for the use of our spectral bound (details in Appendix Section C).

Due to the difficulty of computing $\|\mathbf{J}^{(I)}\|_2$ when the $I^{th}$ layer is a convolution, Singla & Feizi (2020) restrict their analysis to fully connected networks where $\mathbf{J}^{(I)}$ simply equals the $I^{th}$ layer weight matrix. However, using our results in Theorem 1, we can further bound $\mathbf{J}^{(I)}$ and run similar experiments for the convolution layers. Thus, for a 2 layer convolution network, our regularized objective function is given as follows:

$$\min_{\theta} \quad \mathbb{E}_{(\mathbf{x},y)}\big[\ell\big(f_{\theta}(\mathbf{x}),y\big)\big] + \gamma b \, \max_{j}\Big(|\mathbf{W}_{y,j}^{(2)} - \mathbf{W}_{t,j}^{(2)}|\Big)\big(u^{(1)}\big)^2$$

where $\gamma$ is the regularization coefficient, $b$ is a bound on the second-derivative of the activation function ($|\sigma''(.)| \le b$), $(\mathbf{x},y)$'s are the input-label pairs in the training data, and $u^{(I)}$ denotes the bound on the spectral norm for the $I^{th}$ linear (convolution/fully connected) layer. For the convolutional layer, $u^{(1)}$ is again computed as $\sqrt{hw}\min(\|\mathbf{R}\|_2, \|\mathbf{S}\|_2, \|\mathbf{T}\|_2, \|\mathbf{U}\|_2)$ using Theorem 1.

In Table 5, we study the effect of the regularization coefficient $\gamma$ on provable robustness when the network is trained with curvature regularization (Singla & Feizi, 2020). We use a 2 layer convolutional neural network with the tanh (Dugas et al., 2000) activation function and 5 filters in the convolution layer. We observe that the method in Singla & Feizi (2020) coupled with our bound gives significantly higher robustness certificate than CNN-Cert, the previous state of the art (Boopathy et al., 2018).

In Appendix Table 7, we study the effect of the adversarial training method in Singla & Feizi (2020) coupled with our bound on certified robust accuracy. We achieve certified accuracy of $91.25\%$ on MNIST dataset and $29.26\%$ on CIFAR-10 dataset. These results further highlight the usefulness of our spectral bound on convolution layers in improving the provable robustness of the networks against adversarial attacks.

| $\gamma$ | MNIST | | | | |
|---|---|---|---|---|---|
| | Standard Accuracy | Certified Robust Accuracy | CNN-Cert | CRT + Our bound | Certificate Improvement (Percentage %) |
| 0 | 98.35% | 0.0% | 0.1503 | **0.1770** | 17.76% |
| 0.01 | 94.85% | 75.26% | 0.2135 | **0.8427** | 294.70% |
| 0.02 | 93.18% | 74.42% | 0.2378 | **0.9048** | 280.49% |
| 0.03 | 91.97% | 72.89% | 0.2547 | **0.9162** | 259.71% |

Table 5: Comparison between robustness certificates computed using CNN-Cert (Boopathy et al., 2018) and the method proposed in Singla & Feizi (2020) coupled with our spectral bound, for different values of $\gamma$ for a single hidden layer convolutional neural network with tanh activation function. Certified Robust Accuracy is computed as the fraction of correctly classified samples with robustness-certificate (computed using Singla & Feizi (2020)) greater than 0.5.

## 5 CONCLUSION

In this paper, we derive four efficient and differentiable bounds on the spectral norm of convolution layer and take their minimum as our single tight spectral bound. This bound significantly improves over the popular heuristic method of Miyato et al. (2017; 2018), for which we provide the first provable guarantee. Compared to the exact methods of Sedghi et al. (2019) and Ryu et al. (2019), our bound is significantly more efficient to compute, making it amendable to be used in large-scale problems. Over various filter sizes, we empirically observe that the gap between our bound and the true spectral norm is small. Using experiments on MNIST and CIFAR-10, we demonstrate the usefulness of our spectral bound in enhancing generalization as well as provable adversarial robustness of convolutional classifiers.

## 6 ACKNOWLEDGEMENTS

This project was supported in part by NSF CAREER AWARD 1942230, HR001119S0026, HR00112090132, NIST 60NANB20D134 and Simons Fellowship on "Foundations of Deep Learning."

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

# Appendix

## A  NOTATION

For a vector $\mathbf{v}$, we use $\mathbf{v}_j$ to denote the element in the $j^{th}$ position of the vector. We use $\mathbf{A}_{j,:}$ to denote the $j^{th}$ row of the matrix $\mathbf{A}$, $\mathbf{A}_{:,k}$ to denote the $k^{th}$ column of the matrix $\mathbf{A}$. We assume both $\mathbf{A}_{j,:}$ and $\mathbf{A}_{:,k}$ to be column vectors (thus $\mathbf{A}_{j,:}$ is constructed by taking the transpose of $j^{th}$ row of $\mathbf{A}$). $\mathbf{A}_{j,k}$ denotes the element in $j^{th}$ row and $k^{th}$ column of $\mathbf{A}$. $\mathbf{A}_{j,:k}$ and $\mathbf{A}_{:j,k}$ denote the vectors containing the first $k$ elements of the $j^{th}$ row and first $j$ elements of $k^{th}$ column, respectively. $\mathbf{A}_{:j,:k}$ denotes the matrix containing the first $j$ rows and $k$ columns of $\mathbf{A}$:

$$\mathbf{A}_{j,:k} = \begin{bmatrix} \mathbf{A}_{j,0} \\ \mathbf{A}_{j,1} \\ \vdots \\ \mathbf{A}_{j,k-1} \end{bmatrix}, \quad \mathbf{A}_{:j,k} = \begin{bmatrix} \mathbf{A}_{0,k} \\ \mathbf{A}_{1,k} \\ \vdots \\ \mathbf{A}_{j-1,k} \end{bmatrix}, \quad \mathbf{A}_{:j,:k} = \begin{bmatrix} \mathbf{A}_{0,0} & \mathbf{A}_{0,1} & \cdots & \mathbf{A}_{0,k-1} \\ \mathbf{A}_{1,0} & \mathbf{A}_{1,1} & \cdots & \mathbf{A}_{1,k-1} \\ \vdots & \vdots & \ddots & \vdots \\ \mathbf{A}_{j-1,0} & \mathbf{A}_{j-1,1} & \cdots & \mathbf{A}_{j-1,k-1} \end{bmatrix}$$

The same rules can be directly extended to higher order tensors.

For $n \in \mathbb{N}$, we use $[n]$ to denote the set $\{0, \ldots, n\}$ and $[p, q]$ $(p < q)$ to denote the set $\{p, p+1 \ldots, q\}$. We will index the rows and columns of matrices using elements of $[n]$, i.e. numbering from 0. Addition of row and column indices will be done mod $n$ unless otherwise indicated. For a matrix

$\mathbf{A} \in \mathbb{R}^{q \times r}$ and a tensor $\mathbf{B} \in \mathbb{R}^{p \times q \times r}$, $vec(\mathbf{A})$ denotes the vector constructed by stacking the rows of $\mathbf{A}$ and $vec(\mathbf{B})$ denotes the vector constructed by stacking the vectors $vec(\mathbf{B}_{j,:,:})$, $j \in [p-1]$:

$$vec(\mathbf{A}) = \begin{bmatrix} \mathbf{A}_{0,:} \\ \mathbf{A}_{1,:} \\ \vdots \\ \mathbf{A}_{q-1,:} \end{bmatrix}, \qquad vec(\mathbf{B}) = \begin{bmatrix} vec(\mathbf{B}_{0,:,:}) \\ vec(\mathbf{B}_{1,:,:}) \\ \vdots \\ vec(\mathbf{B}_{p-1,:,:}) \end{bmatrix}$$

For a given vector $\mathbf{v} \in \mathbb{R}^n$, $circ(\mathbf{v})$ denotes the $n \times n$ circulant matrix constructed from $\mathbf{v}$ i.e rows of $circ(\mathbf{v})$ are circular shifts of $\mathbf{v}$. For a matrix $\mathbf{A} \in \mathbb{R}^{n \times n}$, $circ(\mathbf{A})$ denotes the $n^2 \times n^2$ doubly block circulant matrix constructed from $\mathbf{A}$, i.e each $n \times n$ block of $circ(\mathbf{A})$ is a circulant matrix constructed from the rows $\mathbf{A}_{j,:}$, $j \in [n-1]$:

$$circ(\mathbf{v}) = \begin{bmatrix} \mathbf{v}_0 & \mathbf{v}_1 & \cdots & \mathbf{v}_{n-1} \\ \mathbf{v}_{n-1} & \mathbf{v}_0 & \cdots & \mathbf{v}_{n-2} \\ \vdots & \vdots & \ddots & \vdots \\ \mathbf{v}_1 & \mathbf{v}_2 & \cdots & \mathbf{v}_0 \end{bmatrix}, \qquad circ(\mathbf{A}) = \begin{bmatrix} circ(\mathbf{A}_{0,:}) & \cdots & circ(\mathbf{A}_{n-1,:}) \\ circ(\mathbf{A}_{n-1,:}) & \cdots & circ(\mathbf{A}_{n-2,:}) \\ \vdots & \ddots & \vdots \\ circ(\mathbf{A}_{1,:}) & \cdots & circ(\mathbf{A}_{0,:}) \end{bmatrix}$$

We use $\mathbf{F}$ to denote the Fourier matrix of dimensions $n \times n$, i.e $\mathbf{F}_{j,k} = \omega^{jk}$, $\omega = e^{-2\pi i/n}$, $i^2 = -1$. For a matrix $\mathbf{A}$, $\sigma(\mathbf{A})$ denotes the set of singular values of $\mathbf{A}$. $\sigma_{max}(\mathbf{A})$ and $\sigma_{min}(\mathbf{A})$ denotes the largest and smallest singular values of $\mathbf{A}$ respectively. $\mathbf{A} \otimes \mathbf{B}$ denotes the kronecker product of $\mathbf{A}$ and $\mathbf{B}$. We use $\mathbf{A} \odot \mathbf{B}$ to denote the hadamard product between two matrices (or vectors) of the same size. We use $\mathbf{I}_n$ to denote the identity matrix of dimensions $n \times n$.

We use the following notation for a convolutional neural network. $L$ denotes the number of layers and $\phi$ denotes the activation $\phi$. For an input $\mathbf{x}$, we use $\mathbf{z}^{(I)}(\mathbf{x}) \in \mathbf{R}^{N_I}$ and $\mathbf{a}^{(I)}(\mathbf{x}) \in \mathbf{R}^{N_I}$ to denote the raw (*before* applying the activation $\phi$) and activated (*after* applying the activation function) neurons in the $I^{th}$ hidden layer of the network, respectively. Thus $\mathbf{a}^{(0)}$ denotes the input image $\mathbf{x}$. To simplify notation and when no confusion arises, we make the dependency of $\mathbf{z}^{(I)}$ and $\mathbf{a}^{(I)}$ to $\mathbf{x}$ implicit. $\phi'(\mathbf{z}^{(I)})$ and $\phi''(\mathbf{z}^{(I)})$ denotes the elementwise derivative and double derivative of $\phi$ at $\mathbf{z}^{(1)}$. $\mathbf{W}^{(I)}$ denotes the weights for the $I^{th}$ layer i.e $\mathbf{W}^{(I)}$ will be a tensor for a convolution layer and a matrix for a fully connected layer. $\mathbf{J}^{(I)}$ denotes the jacobian matrix of $vec(\mathbf{z}^{(1)})$ with respect to the input $\mathbf{x}$. $\theta$ denotes the neural network parameters. $f_\theta(\mathbf{x})$ denotes the softmax probabilities output by the network for an input $\mathbf{x}$. For an input $\mathbf{x}$ and label $y$, the cross entropy loss is denoted by $\ell(f_\theta(\mathbf{x}), y)$.

## B    SEDGHI ET AL'S METHOD

Consider an input $\mathbf{X} \in \mathbb{R}^{c_{in} \times n \times n}$ and a convolution filter $\mathbf{L} \in \mathbb{R}^{c_{out} \times c_{in} \times h \times w}$ to $\mathbf{X}$ such that $n > \max(h, w)$. Using $\mathbf{L}$, we construct a new filter $\mathbf{K} \in \mathbb{R}^{c_{out} \times c_{in} \times n \times n}$ by padding with zeroes:

$$\mathbf{K}_{c,d,k,l} = \left\{ \begin{array}{ll} \mathbf{L}_{c,d,k,l}, & k \in [h-1], \; l \in [w-1] \\ 0, & \text{otherwise} \end{array} \right\}$$

The Jacobian matrix $\mathbf{J}$ of dimensions $c_{out} n^2 \times c_{in} n^2$ is given as follows:

$$\mathbf{J} = \begin{bmatrix} \mathbf{B}^{(0,0)} & \mathbf{B}^{(0,1)} & \cdots & \mathbf{B}^{(0,c_{in}-1)} \\ \mathbf{B}^{(1,0)} & \mathbf{B}^{(1,1)} & \cdots & \mathbf{B}^{(1,c_{in}-1)} \\ \vdots & \vdots & \ddots & \vdots \\ \mathbf{B}^{(c_{out}-1,0)} & \mathbf{B}^{(c_{out}-1,1)} & \cdots & \mathbf{B}^{(c_{out}-1,c_{in}-1)} \end{bmatrix}$$

where $\mathbf{B}^{(c,d)}$ is given as follows:

$$\mathbf{B}^{(c,d)} = circ(\mathbf{K}_{c,d,:,:})$$

By inspection we can see that:

$$vec(\mathbf{Y}) = \mathbf{J} vec(\mathbf{X})$$

From Sedghi et al. (2019), we know that singular values of $\mathbf{J}$ are given by:

$$\sigma(\mathbf{J}) = \bigcup_{j \in [n-1], k \in [n-1]} \sigma(\mathbf{G}^{(j,k)})$$

where each $\mathbf{G}^{(j,k)}$ is a matrix of dimensions $c_{out} \times c_{in}$ and each element of $\mathbf{G}^{(j,k)}$ is given by:

$$\mathbf{G}^{(j,k)}_{c,d} = (\mathbf{F}^T \mathbf{K}_{c,d,:,:} \mathbf{F})_{j,k}, \qquad c \in [c_{out} - 1], \, d \in [c_{in} - 1]$$

The largest singular value of $\mathbf{J}$ i.e $\|\mathbf{J}\|_2$ can be computed as follows:

$$\|\mathbf{J}\|_2 = \sigma_{\max}(\mathbf{J}) = \max_{j \in [n-1], k \in [n-1]} \sigma_{\max}(\mathbf{G}^{(j,k)})$$

## C  SECOND-ORDER ROBUSTNESS

### C.1  ROBUSTNESS CERTIFICATION

Given an input $\mathbf{x}^{(0)} \in \mathbb{R}^D$ and a binary classifier $f$ (here $f : \mathbb{R}^D \to \mathbb{R}$ is differentiable everywhere), our goal is to find a lower bound to the decision boundary $f(\mathbf{x}) = 0$. The $primal$ is given as follows:

$$p^*_{cert} = \min_{\mathbf{x}} \max_{\eta} \left[ \frac{1}{2} \|\mathbf{x} - \mathbf{x}^{(0)}\|_2^2 + \eta f(\mathbf{x}) \right]$$

The $dual$ of the above optimization is given as follows:

$$d^*_{cert} = \max_{\eta} \min_{\mathbf{x}} \left[ \frac{1}{2} \|\mathbf{x} - \mathbf{x}^{(0)}\|_2^2 + \eta f(\mathbf{x}) \right]$$

The inner minimization can be solved exactly when the function $1/2 \|\mathbf{x} - \mathbf{x}^{(0)}\|_2^2 + \eta f(\mathbf{x})$ is strictly convex, i.e has a positive-definite hessian. The same condition can be written as:

$$\nabla^2_{\mathbf{x}} \left[ \frac{1}{2} \|\mathbf{x} - \mathbf{x}^{(0)}\|_2^2 + \eta f(\mathbf{x}) \right] = \mathbf{I} + \eta \nabla^2_{\mathbf{x}} f \succcurlyeq 0$$

It is easy to verify that if the eigenvalues of the hessian of $f$ i.e $\nabla^2_{\mathbf{x}} f$ are bounded between $m$ and $M$ for all $\mathbf{x} \in \mathbb{R}^D$, i.e:

$$m\mathbf{I} \preccurlyeq \nabla^2_{\mathbf{x}} f \preccurlyeq M\mathbf{I}, \qquad \forall \, \mathbf{x} \in \mathbb{R}^D$$

The hessian $\mathbf{I} + \eta \nabla^2_{\mathbf{x}} f \succcurlyeq 0$ (i.e is positive definite) for $-1/M < \eta < -1/m$. We refer an interested reader to Singla & Feizi (2020) for more details and a formal proof of the above theorem.
Thus the inner minimization can be solved exactly for these set of values resulting in the following optimization:

$$q^*_{cert} = \max_{-1/M \leq \eta \leq -1/m} \min_{\mathbf{x}} \left[ \frac{1}{2} \|\mathbf{x} - \mathbf{x}^{(0)}\|_2^2 + \eta f(\mathbf{x}) \right]$$

Note that $q^*_{cert}$ can be solved exactly using convex optimization. Thus, we get the following chain of inequalities:

$$q^*_{cert} \leq d^*_{cert} \leq p^*_{cert}$$

Thus solving $q^*_{cert}$ gives a robustness certificate. The resulting certificate is called *Curvature-based Robustness Certificate*.

### C.2  ADVERSARIAL TRAINING WITH CURVATURE REGULARIZATION

Let $K(\theta, y, t)$ denote the upper bound on the magnitude of curvature values for label $y$ and target $t$ (computed using Singla & Feizi (2020)).
The objective function for training with curvature regularization is given by:

$$\min_{\theta} \quad \mathbb{E}_{(\mathbf{x},y)} \big[ \ell \big( f_\theta(\mathbf{x}), y \big) \big] + \gamma K(\theta, y, t)$$

The objective function for adversarial training with curvature regularization is given by:

$$\min_{\theta} \quad \mathbb{E}_{(\mathbf{x},y)} \big[ \ell \big( f_\theta(\mathbf{x}^{(attack)}), y \big) \big] + \gamma K(\theta, y, t)$$

$\mathbf{x}^{(attack)}$ is computed by doing an adversarial attack on the input $\mathbf{x}$ either using $l_2$ bounded PGD or *Curvature-based Attack Optimization* defined in Singla & Feizi (2020). When *Curvature-based Attack Optimization* is used for adversarial attack, the resulting attack is called *Curvature-based Robust Training* (CRT).

## D  DIFFICULTY OF COMPUTING THE GRADIENT OF CONVOLUTION

Consider a 1D convolution filter of size 3 given by $([1, 2, -1])$ applied to an array of size 5. The corresponding jacobian matrix is given by:

$$\mathbf{J} = \begin{bmatrix} 1 & 2 & -1 & 0 & 0 \\ 0 & 1 & 2 & -1 & 0 \\ 0 & 0 & 1 & 2 & -1 \\ -1 & 0 & 0 & 1 & 2 \\ 2 & -1 & 0 & 0 & 1 \end{bmatrix}$$

The largest singular value $(\|\mathbf{J}\|_2)$, the left singular vector $(\mathbf{u})$ and the right singular vector $(\mathbf{v})$ is given by:

$$\|\mathbf{J}\|_2 = 2.76008, \qquad \mathbf{u} = \begin{bmatrix} -0.55614 \\ 0.11457 \\ 0.62695 \\ 0.27290 \\ -0.45829 \end{bmatrix}, \qquad \mathbf{v} = \begin{bmatrix} -0.63245 \\ -0.19544 \\ 0.51167 \\ 0.51167 \\ -0.19544 \end{bmatrix}$$

The gradient $(\nabla_\mathbf{J} \|\mathbf{J}\|_2)$ is given as follows:

$$\nabla_\mathbf{J} \|\mathbf{J}\|_2 = \begin{bmatrix} 0.35174 & 0.10870 & -0.28456 & -0.28456 & 0.10870 \\ -0.07246 & 0.02239 & 0.05862 & 0.05862 & -0.02239 \\ -0.39652 & -0.12253 & 0.32079 & 0.32079 & -0.12253 \\ -0.17260 & -0.05333 & 0.13964 & 0.13964 & -0.05334 \\ 0.28984 & 0.08957 & -0.23449 & -0.23449 & 0.08957 \end{bmatrix}$$

Clearly the gradient contains non-zero elements in elements of $\mathbf{J}$ that are always zero and unequal gradient values at elements of $\mathbf{J}$ that are always equal.

## E  PROOF OF THEOREM 1

*Proof.* Consider an input $\mathbf{X} \in \mathbb{R}^{c_{in} \times n \times n}$ and a convolution filter $\mathbf{L} \in \mathbb{R}^{c_{out} \times c_{in} \times h \times w}$ to $\mathbf{X}$ such that $n > \max(h, w)$. Using $\mathbf{L}$, we construct a new filter $\mathbf{K} \in \mathbb{R}^{c_{out} \times c_{in} \times n \times n}$ by padding with zeroes:

$$\mathbf{K}_{c,d,k,l} = \begin{cases} \mathbf{L}_{c,d,k,l}, & k \in [h-1], \ l \in [w-1] \\ 0, & \text{otherwise} \end{cases} \tag{2}$$

The output $\mathbf{Y} \in \mathbb{R}^{c_{out} \times n \times n}$ of the convolution operation is given by:

$$\mathbf{Y}_{c,r,s} = \sum_{d=0}^{c_{in}-1} \sum_{k=0}^{n-1} \sum_{l=0}^{n-1} \mathbf{X}_{d,r+k,s+l} \mathbf{K}_{c,d,k,l}$$

We construct a matrix $\mathbf{J}$ of dimensions $c_{out} n^2 \times c_{in} n^2$ as follows:

$$\mathbf{J} = \begin{bmatrix} \mathbf{B}^{(0,0)} & \mathbf{B}^{(0,1)} & \cdots & \mathbf{B}^{(0,c_{in}-1)} \\ \mathbf{B}^{(1,0)} & \mathbf{B}^{(1,1)} & \cdots & \mathbf{B}^{(1,c_{in}-1)} \\ \vdots & \vdots & \ddots & \vdots \\ \mathbf{B}^{(c_{out}-1,0)} & \mathbf{B}^{(c_{out}-1,1)} & \cdots & \mathbf{B}^{(c_{out}-1,c_{in}-1)} \end{bmatrix}$$

where $\mathbf{B}^{(c,d)}$ is given as follows:

$$\mathbf{B}^{(c,d)} = circ(\mathbf{K}_{c,d,:,:})$$

By inspection we can see that:

$$vec(\mathbf{Y}) = \mathbf{J} vec(\mathbf{X})$$

This directly implies that $\mathbf{J}$ is the jacobian of $vec(\mathbf{Y})$ with respect to $vec(\mathbf{X})$ and our goal is to find a differentiable upper bound on the largest singular value of $\mathbf{J}$.

From Sedghi et al. (2019), we know that singular values of $\mathbf{J}$ are given by:

$$\sigma(\mathbf{J}) = \bigcup_{j \in [n-1], k \in [n-1]} \sigma(\mathbf{G}^{(j,k)}) \tag{3}$$

where each $\mathbf{G}^{(j,k)}$ is a matrix of dimensions $c_{out} \times c_{in}$ and each element of $\mathbf{G}^{(j,k)}$ is given by:

$$\mathbf{G}_{c,d}^{(j,k)} = (\mathbf{F}^T \mathbf{K}_{c,d,:,:} \mathbf{F})_{j,k}, \qquad c \in [c_{out} - 1], \; d \in [c_{in} - 1] \tag{4}$$

Using equation 3, we can directly observe that a differentiable upper bound over the largest singular value of $\mathbf{G}^{(j,k)}$ that is independent of $j$ and $k$ will give an desired upper bound. We will next derive the same.

Using equation 4, we can rewrite $\mathbf{G}_{c,d}^{(j,k)}$ as:

$$\mathbf{G}_{c,d}^{(j,k)} = (\mathbf{F}^T \mathbf{K}_{c,d,:,:} \mathbf{F})_{j,k} = (\mathbf{F}_{:,j})^T \mathbf{K}_{c,d,:,:} \mathbf{F}_{:,k} \tag{5}$$

Using equation 2, we know that $\mathbf{K}_{c,d,:,:}$ is a sparse matrix of size $n \times n$ with only the top-left block of size $h \times w$ that is non-zero. We take the first $h$ rows and $w$ columns of $\mathbf{K}_{c,d,:,:}$ i.e $\mathbf{K}_{c,d,:h,:w}$, first $h$ elements of $\mathbf{F}_{:,j}$ i.e $\mathbf{F}_{:h,j}$ and first $w$ elements of $\mathbf{F}_{:,k}$ i.e $\mathbf{F}_{:w,k}$. We have the following set of equalities:

$$\begin{aligned}
\mathbf{G}_{c,d}^{(j,k)} &= (\mathbf{F}_{:,j})^T \mathbf{K}_{c,d,:,:} \mathbf{F}_{:,k} \\
&= (\mathbf{F}_{:h,j})^T \mathbf{K}_{c,d,:h,:w} \mathbf{F}_{:w,k} \\
&= (\mathbf{F}_{:h,j})^T \mathbf{L}_{c,d,:,:} \mathbf{F}_{:w,k}
\end{aligned} \tag{6}$$

Thus, $\mathbf{G}^{(j,k)}$ can be written as follows:

$$\begin{aligned}
\mathbf{G}_{c,d}^{(j,k)} &= \sum_{l=0}^{h-1} \sum_{m=0}^{w-1} \mathbf{F}_{l,j} \mathbf{L}_{c,d,l,m} \mathbf{F}_{m,k} \\
\mathbf{G}^{(j,k)} &= \sum_{l=0}^{h-1} \sum_{m=0}^{w-1} \mathbf{F}_{l,j} \mathbf{L}_{:,:,l,m} \mathbf{F}_{m,k}
\end{aligned} \tag{7}$$

Now consider a block matrix $\mathbf{R}$ of dimensions $c_{out}h \times c_{in}w$ given by:

$$\mathbf{R} = \begin{bmatrix}
\mathbf{L}_{:,:,0,0} & \mathbf{L}_{:,:,0,1} & \cdots & \mathbf{L}_{:,:,0,w-1} \\
\mathbf{L}_{:,:,1,0} & \mathbf{L}_{:,:,1,1} & \cdots & \mathbf{L}_{:,:,1,w-1} \\
\vdots & \vdots & \ddots & \vdots \\
\mathbf{L}_{:,:,h-1,0} & \mathbf{L}_{:,:,h-1,1} & \cdots & \mathbf{L}_{:,:,h-1,w-1}
\end{bmatrix} \tag{8}$$

Thus the block in $l^{th}$ row and $m^{th}$ column of $\mathbf{R}$ is the $c_{out} \times c_{in}$ matrix $\mathbf{L}_{:,:,l,m}$.

Consider two matrices: $(\mathbf{F}_{:h,j})^T \otimes \mathbf{I}_{c_{out}}$ (of dimensions $c_{out} \times c_{out}h$) and $\mathbf{F}_{:w,k} \otimes \mathbf{I}_{c_{in}}$ (of dimensions $c_{in}w \times c_{in}$).

Using equation 7 and equation 8, we can see that:

$$\mathbf{G}^{(j,k)} = \left( (\mathbf{F}_{:h,j})^T \otimes \mathbf{I}_{c_{out}} \right) \mathbf{R} \left( \mathbf{F}_{:w,k} \otimes \mathbf{I}_{c_{in}} \right) \tag{9}$$

Taking the spectral norm of both sides:

$$\begin{aligned}
\left\| \mathbf{G}^{(j,k)} \right\|_2 &= \left\| \left( (\mathbf{F}_{:h,j})^T \otimes \mathbf{I}_{c_{out}} \right) \mathbf{R} \left( \mathbf{F}_{:w,k} \otimes \mathbf{I}_{c_{in}} \right) \right\|_2 \\
\left\| \mathbf{G}^{(j,k)} \right\|_2 &\leq \left\| \left( (\mathbf{F}_{:h,j})^T \otimes \mathbf{I}_{c_{out}} \right) \right\|_2 \left\| \mathbf{R} \right\|_2 \left\| \left( \mathbf{F}_{:w,k} \otimes \mathbf{I}_{c_{in}} \right) \right\|_2
\end{aligned}$$

Using $\|\mathbf{A} \otimes \mathbf{B}\|_2 = \|\mathbf{A}\|_2 \|\mathbf{B}\|_2$ and since both $\|\mathbf{I}_{c_{out}}\|_2$ and $\|\mathbf{I}_{c_{in}}\|_2$ are 1:

$$\left\| \mathbf{G}^{(j,k)} \right\|_2 \leq \|\mathbf{F}_{:h,j}\|_2 \|\mathbf{R}\|_2 \|\mathbf{F}_{:w,k}\|_2$$

Further note that since $\mathbf{F}_{j,k} = \omega^{jk}$, we have $\|\mathbf{F}_{:h,j}\|_2 = \sqrt{h}$ and $\|\mathbf{F}_{:w,k}\|_2 = \sqrt{w}$.

$$\left\| \mathbf{G}^{(j,k)} \right\|_2 \leq \sqrt{hw} \, \|\mathbf{R}\|_2$$

**Alternative inequality (1)**: Note that equation 6 can also be written by taking the transpose of the scalar $(\mathbf{F}_{:h,j})^T \mathbf{L}_{:,:,c,d} \mathbf{F}_{:w,k}$:

$$\mathbf{G}_{c,d}^{(j,k)} = (\mathbf{F}_{:h,j})^T \mathbf{L}_{c,d,:,:} \mathbf{F}_{:w,k} = (\mathbf{F}_{:w,k})^T \left( \mathbf{L}_{c,d,:,:} \right)^T \mathbf{F}_{:h,j}$$

Thus, $\mathbf{G}^{(j,k)}$ can alternatively be written as follows:

$$\mathbf{G}_{c,d}^{(j,k)} = \sum_{l=0}^{w-1} \sum_{m=0}^{h-1} \mathbf{F}_{l,k} \mathbf{L}_{c,d,m,l} \mathbf{F}_{m,j}$$

$$\mathbf{G}^{(j,k)} = \sum_{l=0}^{w-1} \sum_{m=0}^{h-1} \mathbf{F}_{l,k} \mathbf{L}_{:,:,m,l} \mathbf{F}_{m,j} \tag{10}$$

Now consider a block matrix $\mathbf{S}$ of dimensions $c_{out} w \times c_{in} h$ given by:

$$\mathbf{S} = \begin{bmatrix} \mathbf{L}_{:,:,0,0} & \mathbf{L}_{:,:,1,0} & \cdots & \mathbf{L}_{:,:,h-1,0} \\ \mathbf{L}_{:,:,0,1} & \mathbf{L}_{:,:,1,1} & \cdots & \mathbf{L}_{:,:,h-1,1} \\ \vdots & \vdots & \ddots & \vdots \\ \mathbf{L}_{:,:,0,w-1} & \mathbf{L}_{:,:,1,w-1} & \cdots & \mathbf{L}_{:,:,h-1,w-1} \end{bmatrix} \tag{11}$$

For $\mathbf{S}$ the block in $l^{th}$ row and $m^{th}$ column of $\mathbf{R}$ is the $c_{out} \times c_{in}$ matrix $\mathbf{L}_{:,:,m,l}$.
Consider two matrices: $(\mathbf{F}_{:w,k})^T \otimes \mathbf{I}_{c_{out}}$ (of dimensions $c_{out} \times c_{out} w$) and $\mathbf{F}_{:h,j} \otimes \mathbf{I}_{c_{in}}$ (of dimensions $c_{in} h \times c_{in}$).
Using equation 10 and equation 11, we again have:

$$\mathbf{G}^{(j,k)} = \left( (\mathbf{F}_{:w,k})^T \otimes \mathbf{I}_{c_{out}} \right) \mathbf{S} \left( \mathbf{F}_{:h,j} \otimes \mathbf{I}_{c_{in}} \right) \tag{12}$$

Taking the spectral norm of both sides and using the same procedure that we used for $\mathbf{R}$, we get the following inequality:

$$\left\| \mathbf{G}^{(j,k)} \right\|_2 \leq \sqrt{hw} \left\| \mathbf{S} \right\|_2$$

**Alternative inequality (2)**: Using equation 7, $\mathbf{G}^{(j,k)}$ can alternatively be written as follows:

$$\mathbf{G}_{c,d}^{(j,k)} = \sum_{l=0}^{h-1} \sum_{m=0}^{w-1} \mathbf{L}_{c,d,l,m} \mathbf{F}_{l,j} \mathbf{F}_{m,k}$$

$$\mathbf{G}^{(j,k)} = \sum_{l=0}^{h-1} \sum_{m=0}^{w-1} \mathbf{L}_{:,:,l,m} \mathbf{F}_{l,j} \mathbf{F}_{m,k} \tag{13}$$

Now consider a block matrix $\mathbf{T}$ of dimensions $c_{out} \times c_{in} hw$ given by:

$$\mathbf{T} = \begin{bmatrix} \mathbf{A}_0 & \mathbf{A}_1 & \cdots & \mathbf{A}_{h-1} \end{bmatrix} \tag{14}$$

where each block $A_l$ is a matrix of dimensions $c_{out} \times c_{in} w$ given as follows:

$$\mathbf{A}_l = \begin{bmatrix} \mathbf{L}_{:,:,l,0} & \mathbf{L}_{:,:,l,1} & \cdots & \mathbf{L}_{:,:,l,w-1} \end{bmatrix} \tag{15}$$

For $\mathbf{T}$, the block in $l^{th}$ column is the $c_{out} \times c_{in} w$ matrix $\mathbf{A}_l$. For $\mathbf{A}_l$, the block in the $m^{th}$ column is the $c_{out} \times c_{in}$ matrix $\mathbf{L}_{:,:,l,m}$.
Consider the matrix: $\mathbf{F}_{:w,k} \otimes \mathbf{F}_{:h,j} \otimes \mathbf{I}_{c_{in}}$ (of dimensions $c_{in} hw \times c_{in}$).
Using equation 13, equation 14 and equation 15, we again have:

$$\mathbf{G}^{(j,k)} = \mathbf{T} \left( \mathbf{F}_{:h,j} \otimes \mathbf{F}_{:w,k} \otimes \mathbf{I}_{c_{in}} \right) \tag{16}$$

Taking the spectral norm of both sides and using the same procedure that we used for $\mathbf{R}$ and $\mathbf{S}$, we get the following inequality:

$$\left\| \mathbf{G}^{(j,k)} \right\|_2 \leq \sqrt{hw} \left\| \mathbf{T} \right\|_2$$

**Alternative inequality (3)**: Using equation 7, $\mathbf{G}^{(j,k)}$ can alternatively be written as follows:

$$\mathbf{G}_{c,d}^{(j,k)} = \sum_{l=0}^{h-1} \sum_{m=0}^{w-1} \mathbf{F}_{l,j} \mathbf{F}_{m,k} \mathbf{L}_{c,d,l,m}$$

$$\mathbf{G}^{(j,k)} = \sum_{l=0}^{h-1} \sum_{m=0}^{w-1} \mathbf{F}_{l,j} \mathbf{F}_{m,k} \mathbf{L}_{:,:,l,m} \tag{17}$$

Now consider a block matrix $\mathbf{U}$ of dimensions $c_{out}hw \times c_{in}$ given by:

$$\mathbf{U} = \begin{bmatrix} \mathbf{B}_0 \\ \mathbf{B}_1 \\ \vdots \\ \mathbf{B}_{h-1} \end{bmatrix} \tag{18}$$

where each block $\mathbf{B}_l$ is a matrix of dimensions $c_{out}w \times c_{in}$ given as follows:

$$\mathbf{B}_l = \begin{bmatrix} \mathbf{L}_{:,:,l,0} \\ \mathbf{L}_{:,:,1,1} \\ \vdots \\ \mathbf{L}_{:,:,l,w-1} \end{bmatrix} \tag{19}$$

For $\mathbf{U}$, the block in $l^{th}$ row is the $c_{out}w \times c_{in}$ matrix $\mathbf{B}_l$. For $\mathbf{B}_l$, the block in the $m^{th}$ row is the $c_{out} \times c_{in}$ matrix $\mathbf{L}_{:,:,l,m}$.

Consider the matrix: $\left( (\mathbf{F}_{:h,j})^T \otimes (\mathbf{F}_{:w,k})^T \otimes \mathbf{I}_{c_{out}} \right)$ (of dimensions $c_{out} \times c_{out}hw$).

Using equation 17, equation 18 and equation 19, we again have:

$$\mathbf{G}^{(j,k)} = \left( (\mathbf{F}_{:h,j})^T \otimes (\mathbf{F}_{:w,k})^T \otimes \mathbf{I}_{c_{out}} \right) \mathbf{U} \tag{20}$$

Taking the spectral norm of both sides and using the same procedure that we used for $\mathbf{R}, \mathbf{S}$ and $\mathbf{T}$, we get the following inequality:

$$\left\| \mathbf{G}^{(j,k)} \right\|_2 \le \sqrt{hw} \left\| \mathbf{U} \right\|_2$$

Taking the minimum of the four bounds i.e $\sqrt{hw} \min( \|\mathbf{R}\|_2, \|\mathbf{S}\|_2, \|\mathbf{T}\|_2, \|\mathbf{U}\|_2)$, we have the stated result. $\qquad \square$

## F    Proof of Lemma 1

*Proof.* When $h = 1$ and $w = 1$, note that in equations 9, 12, 16 and 20, we have:

$$\mathbf{F}_{:h,j} = 1, \qquad \mathbf{F}_{:w,k} = 1$$

This directly implies:

$$\|\mathbf{J}\|_2 = \|\mathbf{R}\|_2 = \|\mathbf{S}\|_2 = \|\mathbf{T}\|_2 = \|\mathbf{U}\|_2$$

$\qquad \square$

## G    Effect of increasing $\beta$ on singular values

In Figure 1, we plot the effect of increasing $\beta$ on the sum of true singular values of the network and sum of our upper bound. We observe that the gap between the two decreases as we increase $\beta$.

In Table 6, we show the effect of increasing $\beta$ on the bound on the largest singular value of each layer.

## H    Effect on certified robust accuracy

| Filter shape | $\beta$ values | | | | | | | |
|---|---|---|---|---|---|---|---|---|
| | 0 | 0.0006 | 0.0008 | 0.001 | 0.0012 | 0.0014 | 0.0016 | 0.0018 |
| $16 \times 3 \times 3 \times 3$ | 37.96 | 11.05 | 9.13 | 8.2 | 7.28 | 8.66 | 7.06 | 5.57 |
| $16 \times 16 \times 3 \times 3$ | 22.23 | 5.89 | 5.08 | 5.26 | 5.19 | 6.17 | 3.41 | 3.27 |
| $16 \times 16 \times 3 \times 3$ | 25.33 | 5.69 | 4.87 | 4.9 | 4.75 | 5.80 | 3.26 | 3.27 |
| $16 \times 16 \times 3 \times 3$ | 25.29 | 5.28 | 4.74 | 4.08 | 2.89 | 6.60 | 3.72 | 2.32 |
| $16 \times 16 \times 3 \times 3$ | 21.66 | 4.99 | 4.36 | 3.64 | 2.64 | 5.81 | 3.61 | 2.25 |
| $16 \times 16 \times 3 \times 3$ | 20.15 | 4.92 | 3.89 | 4.61 | 2.53 | 3.13 | 0.07 | 2.4 |
| $16 \times 16 \times 3 \times 3$ | 16.36 | 4.37 | 3.52 | 4.08 | 2.34 | 2.60 | 0.07 | 2.14 |
| $16 \times 16 \times 3 \times 3$ | 15.39 | 5.39 | 5.65 | 3.15 | 4.1 | 4.72 | 2.31 | 1.69 |
| $16 \times 16 \times 3 \times 3$ | 14.67 | 4.38 | 4.61 | 2.32 | 3.52 | 3.66 | 2.37 | 1.57 |
| $16 \times 16 \times 3 \times 3$ | 19.26 | 7.41 | 6.28 | 4.96 | 3.57 | 0.13 | 5.38 | 3.19 |
| $16 \times 16 \times 3 \times 3$ | 14.60 | 5.77 | 4.84 | 4.01 | 2.68 | 0.09 | 4.31 | 2.64 |
| $32 \times 16 \times 3 \times 3$ | 23.24 | 8.53 | 8.09 | 7.06 | 6.24 | 6.30 | 6.4 | 6.61 |
| $32 \times 32 \times 3 \times 3$ | 22.19 | 9.65 | 9.57 | 7.99 | 7.27 | 7.12 | 7.72 | 7.82 |
| $32 \times 32 \times 3 \times 3$ | 17.97 | 7.61 | 7.52 | 6.32 | 4.72 | 5.25 | 5.06 | 2.57 |
| $32 \times 32 \times 3 \times 3$ | 16.76 | 6.83 | 6.29 | 5.38 | 4.12 | 4.48 | 4.48 | 2.22 |
| $32 \times 32 \times 3 \times 3$ | 17.62 | 8.31 | 6.2 | 6.73 | 4.06 | 5.23 | 5.65 | 3.21 |
| $32 \times 32 \times 3 \times 3$ | 15.7 | 7.24 | 5.23 | 5.5 | 3.42 | 4.45 | 4.73 | 2.8 |
| $32 \times 32 \times 3 \times 3$ | 15.82 | 8.34 | 5.42 | 5.29 | 4.86 | 5.41 | 5.03 | 4.11 |
| $32 \times 32 \times 3 \times 3$ | 15.83 | 6.99 | 4.48 | 4.47 | 4.02 | 4.62 | 4.24 | 3.56 |
| $32 \times 32 \times 3 \times 3$ | 18.46 | 8.1 | 6.03 | 5.31 | 5.91 | 5.64 | 4.83 | 6.24 |
| $32 \times 32 \times 3 \times 3$ | 17.75 | 7.08 | 5.17 | 4.53 | 4.98 | 4.61 | 4.15 | 5.31 |
| $64 \times 32 \times 3 \times 3$ | 23.57 | 12.21 | 10.97 | 10.41 | 10.84 | 10.15 | 9.7 | 10.31 |
| $64 \times 64 \times 3 \times 3$ | 22.97 | 13.2 | 12.23 | 11.72 | 12.55 | 11.36 | 11.14 | 12.39 |
| $64 \times 64 \times 3 \times 3$ | 22.35 | 12.18 | 11.71 | 11.13 | 10.53 | 10.57 | 10.2 | 10.43 |
| $64 \times 64 \times 3 \times 3$ | 22.02 | 11.28 | 10.95 | 10.43 | 9.94 | 9.94 | 9.52 | 9.55 |
| $64 \times 64 \times 3 \times 3$ | 20.91 | 12.09 | 11.7 | 11.07 | 10.47 | 10.85 | 9.87 | 9.81 |
| $64 \times 64 \times 3 \times 3$ | 21.77 | 10.85 | 10.51 | 9.81 | 9.6 | 9.62 | 8.56 | 8.43 |
| $64 \times 64 \times 3 \times 3$ | 19.63 | 11.97 | 11.56 | 10.28 | 10.98 | 10.02 | 9.32 | 8.69 |
| $64 \times 64 \times 3 \times 3$ | 17.61 | 10 | 10.03 | 8.81 | 9.63 | 8.26 | 7.57 | 7.35 |
| $64 \times 64 \times 3 \times 3$ | 16.67 | 10.34 | 11.61 | 10.26 | 10.93 | 8.64 | 8.05 | 8.56 |
| $64 \times 64 \times 3 \times 3$ | 17.81 | 8.16 | 8.73 | 7.68 | 8.18 | 6.61 | 6.07 | 6.49 |
| $10 \times 64$ | 12.43 | 10.79 | 10.55 | 10.53 | 10.11 | 10.14 | 9.64 | 9.39 |

Table 6: Effect of increasing $\beta$ on the bounds ($\sqrt{hw}\|\mathbf{R}\|_2$) of each layer

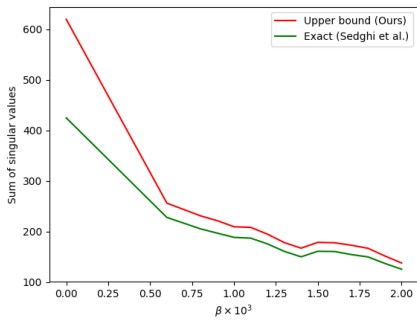

Figure 1: We plot the effect of increasing $\beta$ on the sum of the bounds (computed using our method i.e $\sqrt{hw}\|\mathbf{R}\|_2$) and sum of true spectral norms ($\|\mathbf{J}^{(I)}\|_2$ computed using Sedghi et al. (2019)) for a Resnet-32 neural network trained on CIFAR-10 dataset (without any weight decay). We observe that the gap between the two decreases as we increase $\beta$. In Table 6, we show the effect of increasing $\beta$ on the bounds of individual layers of the same network.)

| $\gamma$ | MNIST | | | CIFAR-10 | | |
|---|---|---|---|---|---|---|
| | Standard Accuracy | Empirical Robust Accuracy | Certified Robust Accuracy | Standard Accuracy | Empirical Robust Accuracy | Certified Robust Accuracy |
| 0 | 98.68% | 87.81% | 0.00% | 56.22% | 14.88% | 0.00% |
| 0.01 | 97.08% | 92.92% | **91.25%** | 53.52% | 31.82% | 17.39% |
| 0.02 | 96.36% | 90.98% | 89.58% | 49.55% | 31.80% | 25.93% |
| 0.03 | 95.54% | 89.99% | 88.75% | 46.56% | 31.98% | **29.26%** |

Table 7: Comparison between certified robust accuracy for different values of the regularization parameter $\gamma$ for a single hidden layer convolutional neural network with softplus activation function. Certified Robust Accuracy is computed as the fraction of correctly classified samples with CRC (Curvature-based Robustness Certificate as defined in Singla & Feizi (2020)) greater than 0.5. Empirical robust accuracy is computed by running $l_2$ bounded PGD (200 steps, step size 0.01). We use convolution layer with 64 filters and stride of size 2.

