# OpenReview forum: "Fantastic Four: Differentiable and Efficient Bounds on Singular Values of Convolution Layers"
_ICLR.cc/2021/Conference — ICLR 2021 Poster_

### Official Review · AnonReviewer4 · 2020-10-21
**Efficient upper bounds for spectral norm of CNNs**

**Rating:** 8
**Confidence:** 5

**Review:**

This paper provides an method for computing an upper bound for the spectral norm of the linear transformation induced by a convolutional layer. An upper bound was first introduced as a heuristic by Miyato et al, but they did not prove any bounds. The authors use the exact computation of singular values of a convolutional layer in Sedghi, Gupta and Long to prove that the Miyato heuristic is indeed an upper bound. They further generalize Miyato's method to find 3 additional heuristics, all of which are proved to be upper bounds, and then show empirically that the minimum of these bounds gives a much tighter bound, often very close to the exact value. The bounds are significantly faster to compute than exact spectral norms, both in complexity and in practice.

The authors show that their bounds can be used for regularization, and produce results similar to the spectral projection method in Sedghi et al on CIFAR-10. They also show that spectral bounds enhance the robustness of CNNs against adversarial attacks - previous papers were unable to compute these spectral norms so remained restricted to fully connected networks. By extending the results of Singla & Feizi to convolutional nets, the authors are able to improve the state of the art significantly.

Strengths:
The upper bounds are formally proved, and the proofs are easy to follow. The upper bounds are easy to compute - the methods overcomes the dependence on n^2 in the exact methods.

Weaknesses:
The upper bounds only bound the spectral norm, and do not provide any information about the remaining singular values. The comparison with Sedghi's CIFAR-10 used a baseline that was weaker, Sedghi et al improved CIFAR-10 accuracy from 93.8 to 94.7. I would like Section 4.3 to be explained a bit better, so that I can understand the significance of the results more clearly.

---

> ### Author Response · Authors · 2020-11-24
> **Rebuttal #4**
>
> Thank you for your positive feedback. In the updated version of the paper, we will include a more detailed comparison with previous methods for the robustness application.

---

### Official Review · AnonReviewer2 · 2020-10-28
**Manuscript is easy to read, but I have a few concerns**

**Rating:** 5
**Confidence:** 4

**Review:**

Summary:
In this paper, an upper bound of the spectral norm of Jacobian of a convolutional layer is presented. The proposed bound enables us to compute the gradient much faster than the exact methods. In several experiments, the tightness of the bound and runtime/accuracy tradeoff are reported.

Pros:
P1. A simple yet practical bound is provided
P2. Code is provided, which is clean and easy to read
P3. The problem is well motivated with a sufficient literature survey, which will stimulate the ICLR community

Cons:
C1. The experiments do not include the seminal prior work (Miyato+ 2018) as a baseline
C2. The terminology of "difficult/easy" for gradient computation is unclear
C3. The technical contributions are relatively marginal


Detailed comments:

Overall, the paper is clearly written, and easy to follow the content.

My concerns:
- [C1] One-step power method proposed by Miyato+ (2017; 2018) is not compared in the experiments. Since the proposed method is a generalization of Miyato's method, it looks like a natural way to make comparisons, e.g. in Section 4.2. Without comparison, the significance of this study would be not fully validated --- if Miyato's method achieves almost the same performance as the proposed method, the practical value would be degraded.
- [C2] What does "easy" or "difficult" mean in terms of gradient computation? Does it refer to numerical stability, computational complexity, or something else? It seems there is no technical definition of them.
- [C3] From (Miyato+ 2018), the main technical contributions are 1. proving that Miyato's method actually optimizes the upper bound of spectral norm up to constant scale, 2. tightening the bound by considering three more different reshaped matrices (R, S, U). In a theoretical sense, both 1 and 2 are meaningful. However, 1 does not contribute to the practical value, because when we use the spectral norm as a regularizer, the constant factor is absorbed in hyperparameter beta in (1). Also, there is no empirical evidence of how 2 improves the performance, as mentioned in [C1].

---

> ### Author Response · Authors · 2020-11-24
> **Rebuttal #2**
>
> 1. We followed the experimental practice of Sedghi et al. (published in ICLR 2018). They show that their method results in improvements over standard training. We show that our method gives similar improvements but does require a significantly less running time, that is independent of input size and hence can be used for large datasets such as Imagenet.
>
> 2. The method of gradient computation for the methods of Sedghi et al. and Ryu et al. was never discussed in their papers. Ryu et al use power iteration to compute the largest singular value of convolution operation. The gradient can be computed using the outer product of the largest singular vectors. But this results in a problem as we discuss in Appendix Section D. For Sedghi et al’s method, the outer product of largest singular value will be a complex matrix of size $c_{out} \times c_{in}$. This needs to be projected to a convolution filter which is real (not complex) and of size $c_{out} x c_{in} \times h \times w$. We discuss these two issues in the introduction of the paper.
>
> 3. In Tables 1 and 3, we show significant improvements over the heuristic of Miyato et al.

---

### Official Review · AnonReviewer3 · 2020-10-28
**Important literature is missing**

**Rating:** 3
**Confidence:** 4

**Review:**

## Summary of the paper

This paper proposes an improved method to calculate an upper bound of the spectral norm of the 2d-convolutional layer. The advantage of the proposed method is fast computation and easy gradient computation. Experiments on MNIST and CIFAR10 show that regularizing spectral norms using the proposed method improves generalization. Additionally, they showed that their method could extend and improve an existing method for computing certified robustness accuracy.

## Review

### Summary

The paper is clearly written and easy to understand. However, the paper misses important literature and also seems to misunderstand existing methods. And I found the proposed method does not have the claimed advantages over existing methods.

### More detailed comments

1.  Misunderstandings of the existing method
If I understand it correctly, Ryu et al. do not directly calculate the singular vectors' outer-product. The spectral norm calculation written in Ryu et al. is differentiable. And thus, you can directly take the gradient of the spectral norm, and major deep learning frameworks efficiently compute its gradients.

2. Missing literature

> this is the first work that derives a provable bound on the spectral norm of convolution filter as a constant factor (dependant on filter sizes, but not filter weights) times the heuristic of Miyato et al. (2018)

It's at least mentioned in Cisse et al. (2017).

Additionally, efficient and differentiable computation of linear operators appeared in the literature repeatedly (Tsuzuku et al. 2018, Scaman & Virmaux 2018). They are not limited to the 2d convolutional layers, but any linear layers implemented using deep learning frameworks. Tsuzuku et al. 2018 also applied the method to improve certified robustness accuracy. I was not convinced that the proposed method has more advantages over these methods.


3. Weak experiments

The experimental results seem noisy, and also the gain by the proposed regularizer looks marginal. Please write the number of each run and their standard deviations. Additionally, I am not convinced that the tighter bound results in better generalization. The proposed method is very similar to Yoshida & Miyato 2017, and it is not intuitive that the proposed method works better than it. Please add appropriate experimental comparisons with Yoshida & Miyato 2017.

## References

Cisse et al. Parseval networks: Improving robustness to adversarial examples. ICML 2017.
Scaman and Virmaux. Lipschitz regularity of deep neural networks: analysis and efficient estimation. NeurIPS 2018.
Tsuzuku et al. Lipschitz-Margin Training: Scalable Certification of Perturbation Invariance for Deep Neural Networks. NeurIPS 2018
Yoshida and Miyato, Spectral norm regularization for improving the generalizability of deep learning. ArXiv, abs/1705.10941, 2017.

---

> ### Author Response · Authors · 2020-11-24
> **Rebuttal #3**
>
> 1. In contributions, on Page 2 (https://arxiv.org/pdf/1905.05406.pdf), Ryu at al make the following statement:
>
> "For the analysis, we assume the denoiser $H_{\sigma}$ satisfies a certain Lipschitz condition, formally defined as Assumption (A). Roughly speaking, the condition corresponds to the denoiser $H_{\sigma}$ being close to the identity map, which is reasonable when the denoising parameter \sigma is small. In particular, we do not assume that $H_{\sigma}$ is nonexpansive or differentiable since most denoisers do not have such properties".
>
> In that paper, they do not assume the denoiser (that they later construct using real normalization) to be differentiable. Moreover, we never see the differentiability of real normalization discussed in the paper.
>
> 2. Cisse et al. (2017) derive this bound only for a 1d convolution while we derive the same for 2d convolution filter. Scaman & Virmaux 2018 derive a bound that is similar to Ryu et al. but requires backpropagation through the convolution layer at every time step while Ryu et al only requires forward computation i.e a convolution operation. We include comparison with the more recent state-of-the-art work (CNN-CERT by Boopathy et al.) and show significant improvements. The other works on certification, Scaman & Virmaux 2018, Tsuzuku et al. 2018 are relatively older works and thus we do not include them in our comparison.
>
> 3. We followed the experimental practice of Sedghi et al. (published in ICLR 2018). They show that their method results in improvements over standard training. We show that our method gives similar improvements but does require a significantly less running time, that is independent of input size and hence can be used for large datasets such as Imagenet.

---

### Official Review · AnonReviewer1 · 2020-10-28
**Intersting development but the evaluation process is not sufficient**

**Rating:** 4
**Confidence:** 4

**Review:**

## Summary

This paper propose to study the Lipschitz constant of convolutional layers and to give an easy to compute and differentiable upper bound. The upper bound is composed of 4 different bounds, based on tensor unfolding of the Jacobian, and taking the min of these 4 values. This upper bound is then used to train networks with spectral norm regularization and compared with network trained with singular value clipping from `Sedghi et al. (2019)`. The proposed bound gives similar performances with much cheaper computational time.


## Overall assessment

- The title is misleading a bit. The bound is not on all the singular values but on the largest one. I would update it to `Fantastic four: differentiable upper bound on the Lipschitz constant of convolutional layers`.
- The derivations for the 4 bounds are very straightforward from the results from `Sedghi et al. (2019)` as it is simply different unfolding of the hollow tensor that is obtained with the Jacobian of the convolution layer in the Fourier domain. The Jacobian is given in the preceding paper so the technical contribution is not very large.
- My main concern with this paper is that the proposed evaluation is not sufficient to show the benefit of the proposed method IMO.
    * First, I don't understand why the author choosed to compare singular value clipping with the spectral norm regularisation. Indeed, computing all the singular value will be much more expensive than computing a bound on the largest one. A proper evaluation with the same regularisation, using for instance the low number of power iterations proposed in `Ryu et al. (2019)` would be more convincing to show the computational benefit as well to compare the potential accuracy loss due to the approximation.
    * Then, the results seems to be provided on single run of the method and not average on multiple random init. Averaging (and giving the standard deviation) would allow to asses the difference in stability of the proposed result.
    * The weight decay is "selected using grid search" but there is no mention of validation set or the score used to select this parameter. This feels like the weight decay has been selected to maximize the test error, which is not a proper method to select such parameter.
- I don't understand why the running time of the proposed method does not change between layer of size 64x64x3x3 and 512x512x3x3 in table.1. As the complexity of the bound is at least linear in the total shape of the filter, the running time should be mutliply by 64, which would give similar runtime as the one in Ryu et al (2019).


## Minor comments, nitpicks and typos

- p.3: `The corresponding Jacobian matrix` -> relative to the input. An introduction of the layer as a function $\phi(x) = w * x$ and the definition of the Jacobian $J = \frac{\partial \phi}{\partial x}(x)$ would probably help in this regard.
- Table.1: this table could probably be reduced to make more space for experiements
- p.4: the notation section introduce many notations that are not used in the core of the paper. This should be trimmmed down to the minimum. For instance, I don't think that $\phi'(z), \phi''(z)$ or the per layer $z^{(I)}, a^{(I)}$ are used in the core of the text.
- p.5: `O(1/p)bound is obtained on the error`: which error are the authors referring to? This should be properly introduced and discuss.
- p.7: In the last paragraph of 4.2, the results are first given in order "no weight decay/ with with decay" and then in reverse order. This is confusing to read and changing the order would improve the readability.
- Sedghi et al. (2018): The paper was pucblished in ICLR 2019, the proper citation is:
```bibtex
@inproceedings{Sedghi2019,
    title={The Singular Values of Convolutional Layers},
    author={Hanie Sedghi and Vineet Gupta and Philip M. Long},
    booktitle={International Conference on Learning Representations},
    year={2019},
    url={https://openreview.net/forum?id=rJevYoA9Fm},
}
```
- Ryu et al. (2019): The paper was published in ICML 2019:
```bibtex
 @InProceedings{pmlr-v97-ryu19a,
    title = {Plug-and-Play Methods Provably Converge with Properly Trained Denoisers},
    author = {Ryu, Ernest and Liu, Jialin and Wang, Sicheng and Chen, Xiaohan and Wang, Zhangyang and Yin, Wotao},
    pages = {5546--5557},
    year = {2019},
    editor = {Kamalika Chaudhuri and Ruslan Salakhutdinov},
    volume = {97},
    series = {Proceedings of Machine Learning Research},
    address = {Long Beach, California, USA},
    month = {09--15 Jun},
    publisher = {PMLR},
    pdf = {http://proceedings.mlr.press/v97/ryu19a/ryu19a.pdf},
    url = {http://proceedings.mlr.press/v97/ryu19a.html}
}
 ```
- Other references should be checked.
- Section D: The complex part here is not to compute the gradient of the convolution (which is simply computed with the correlation) but the shape of the gradient of the spectral norm of the Jacobian. This should be updated.
- p.14: `will give a{n,} desired`.
- p14/15/26: Eq.6/10/13/17 are the same (multiplication is associative and l/m are inverted in eq 10). The repetition is misleading as the reader search for the difference which is simply the ordering of terms. All these bounds would be much easier to derive/read if using tensor notation. Basically, what is done here seems to simply be computing the spectral norms of tensor slices unfolded along different dimensions.

---

> ### Author Response · Authors · 2020-11-24
> **Rebuttal #1**
>
> 1. Paper title: Since our bound is on the largest singular value of a convolution layer, by definition, it also bounds all the singular values.
> 2. Technical contribution: Sedghi et al. also derived an upper bound in the previous version of their paper: https://arxiv.org/abs/1805.10408v1 which was equal to the sum of absolute values of all weights in the convolution filter (and hence significantly different and less tight from our bound). They removed it in the next version (https://arxiv.org/pdf/1805.10408.pdf). Therefore, we do not believe it is fair to say that our result is a straightforward extension of Sedghi et al. Moreover, as we show in Tables 1 and 3, our bound results in significant improvement over the method of Miyato et al highlighting its practical importance.
> 3. We followed the experimental practice of Sedghi et al. (published in ICLR 2018). They show that their method results in improvements over standard training. We show that our method gives similar improvements but does require a significantly less running time, which is independent of input size and hence can be used for large datasets such as Imagenet.
> 4. Weight decay: We used a held out validation set of 5000 examples. We will include the details in the updated version of the paper.
> 5. The computational complexity we derive assumes a sequential computation (not parallel as in modern GPUs). Since matrix multiplication operation in GPU is very fast, we do not see a significant change for different sizes of convolution filters. Moreover, the method of Ryu at al is not independent of input size.
> 6. Thanks for your comments. We will make the suggested edits in the updated version.

---

### Decision · Program_Chairs · 2021-01-07
**Final Decision**

**Decision:**

Accept (Poster)

**Comment:**

The authors provide four rigorous upper bounds on the operator norm of the linear transformation associated with a 2D convolutional layer of a neural network.  One of these is a heuristic proposed in earlier work by Miyato et al, and widely used, so, among other things, their result provides theoretical context for that method which will be of broad interest.  All four of their bounds can be efficiently computed and have easily computed gradients, so they propose using the minimum of the four bounds for various purposes.  Since, for standard architectures, the Lipschitz constant of a network can be bounded above by the product of the operator norms of its layers, there are a variety of applications of differentiable bounds on these operator norms.  They show that their new bound is sometimes much tighter than the bound of Miyato et al, and can be computed much more efficiently than two known methods for exact computation.  The paper is written well, which will facilitate future work building on this work.  The analysis builds on earlier work, but insight was required to obtain the new results;  the fundamental novelty of the mathematical development was confirmed by an expert reviewer.

While they experimentally compared the accuracy of their approximations to those of the method of Miyato, et al, the case for the practical utility of their method would have been stronger if they had shown that their regularizer led to better results for some tasks.  However, I believe that the paper should be accepted purely on the basis of its theoretical contribution, which enhances our understanding of this important topic, and, even if it cannot be directly applied, seems like to inspire practically useful methods in the future.